# The Effect of Fe Addition on the Curie Temperature and Magnetic Entropy of the Gd_45_Co_50_Al_5_ Amorphous Alloy

**DOI:** 10.3390/ma16134571

**Published:** 2023-06-24

**Authors:** Luyi Li, Benzhen Tang, Weijie Fu, Ying Lu, Yunqing Fu, Ding Ding, Lei Xia, Peng Yu

**Affiliations:** 1Chongqing Key Laboratory of Photo-Electric Functional Materials, College of Physics and Electronic Engineering, Chongqing Normal University, Chongqing 401331, China; li_lu_yi@yeah.net (L.L.); fuweijieyxs@yeah.net (W.F.); lu_ying_l@yeah.net (Y.L.); fyunqing@yeah.net (Y.F.); 2Institute of Materials & Laboratory for Microstructure, Shanghai University, Shanghai 200072, China; d.ding@shu.edu.cn (D.D.); xialei@shu.edu.cn (L.X.)

**Keywords:** amorphous alloy, magnetocaloric effect, Curie temperature

## Abstract

The new magnetic refrigeration (MR) technology, which uses the magnetocaloric effect (MCE) of materials for refrigeration, has shown apparent advantages over the compression refrigeration of freon and other gases. Therefore, how to obtain materials with excellent magnetic entropy change near room temperature is of great significance for the realization of MR. In order to achieve high Tc of a Gd-based amorphous alloy, Gd_45_Co_50_Al_5_ amorphous alloy with good room temperature MCE was selected, and a series of Gd_45_Co_50−x_Fe_x_Al_5_ (x = 2, 5, 10) amorphous alloys were prepared by adding Fe instead of Co. In this paper, the effect of Fe addition on the Curie temperature, and the magnetic entropy change in the alloys, were studied thoroughly. The results show that the Curie temperature is increased to 281 K by adding 5% Fe elements, which is mainly related to the enhanced 3*d*-3*d* interaction of transition elements caused by Fe addition, and the maximum value of magnetic entropy change is 3.24 J/(kg·K) under a field of 5 T. The results are expected to provide guidance for further improving the room temperature MCE of Gd-based amorphous alloys.

## 1. Introduction

Nowadays, clean and green energy has been touted in many fields due to its advantages of high efficiency and energy saving. In the field of refrigeration, the traditional refrigerator, which is characterized by compressed fluorocarbons and fluorine refrigeration gases, has been unable to meet the needs in the current era of high-quality development due to the disadvantages such as high energy consumption and environmental destruction [1,2,3,4]. By utilizing the magnetocaloric effect (MCE) of materials, that is, the phenomenon that the ferromagnet or paramagnet generate heat change when stimulated by an external magnetic field during adiabatic process, this new magnetic refrigeration (MR) pattern can realize a heat interaction with the exterior and the refrigeration efficiency can reach up to 60% of the Carnot cycle. Compared with the conventional gas compression refrigeration, MR shows the advantages of being more efficient, energy saving and environmentally friendly, therefore, it is considered to be a very advantageous new field [5,6,7,8].

The realization of MR depends on the MCE of materials [1,2]. Therefore, how to obtain large magnetic refrigeration capacity in a large temperature range, especially in the temperature range of room temperature, so as to obtain as much refrigeration efficiency as possible, is the topic issue regarding the application of MR at room temperature [9]. Among the prepared magnetic refrigerants at present, the materials with second-order magnetic phase transition (SOMPT), which are characterized by a continuous magnetic phase transition process, were considered to be the best choice because of their wide range of magnetic entropy change (−Δ*S_m_*) region. Among these SOMPT MR materials, amorphous alloys (AM) have gained wide attention [10,11]. AM is in an energy metastable state due to its long-range disorder of the atomic arrangement, thus showing many unique performance advantages, such as good machinability, high corrosion resistance and large resistance to reduce eddy current loss [10,11,12,13,14,15,16,17,18,19,20,21,22]. In addition, AM can be formed within a certain composition range and the magnetic properties can be adjusted continuously with the change in composition, so it is possible for us to design a refrigerant with good MCE performance within a certain temperature range.

Since the −Δ*S_m_* of materials are closely related to angular momentum *J*, rare-earth (RE)-based AM usually show excellent −Δ*S_m_* properties [6,7]. However, the Curie temperature (*T_c_*) of RE-based amorphous alloys is generally in a low temperature range, mostly below the cooling temperature of liquid nitrogen [11,12,13,14,15,16,17]. Meanwhile, the AM that are prepared by Gd have the *T_c_* closest to room temperature due to the half-full structure of their 4*f* layer. Among them, the Curie temperature of Gd_50_Co_50_ amorphous alloy reaches 267.5 K, and the peak of magnetic entropy change (−Δ*S_m_^peak^*) and refrigeration capacity (RC = −Δ*S_m_^peak^* × δ*T_FWHM_* means the comprehensive MCE of the material, which can be approximated by the product of the −Δ*S_m_^peak^* and the temperature interval (δ*T_FWHM_*) that corresponds to the full width temperature range at the half value of the −Δ*S_m_^peak^* in the −Δ*S_m_* curve) under a field of 5 T, are approximately 4.6 J/(kg·K) and 685.9 J/kg [18,19]. Afterwards, the *T_c_* of Gd-based AM was successfully raised to near room temperature by the addition of the transition elements, Fe and Ni, among which Gd_50_Co_50−x_Fe_x_ (x = 2, 5) and Co_50_Gd_50−x_(Fe/Ni)_x_ (x = 1, 2, 3) AM showed the highest *T_c_* of the RE-based AM, which can reach 289 K (Gd_50_Co_45_Fe_5_), 289 K (Co_50_Gd_47_Ni_3_) and 297 K (Co_50_Gd_48_Ni_2_), respectively [20,21,22,23]. However, with the increase in Fe and Ni content, the glass-forming ability (GFA) of these alloys deteriorates significantly, which makes the preparation of AM with higher *T_c_* slightly difficult. The addition of Al can further ameliorate this situation. Gd_45_Co_50_Al_5_ AM not only reduces Gd content to 45%, but also further improves the forming capacity of the AM. However, the only issue is that the *T_c_* of Gd_45_Co_50_Al_5_ AM is only 253 K.

Therefore, in order to further obtain Gd-based AM with *T_c_* at room temperature and a better GFA, a series of Gd_45_Co_50−x_Fe_x_Al_5_ (x = 2, 5, 10) AM based on a Gd_45_Co_50_Al_5_ sample were prepared by modulation of the proportion of transition elements through adding Fe instead of Co. The effect of Fe addition on the magnetic properties (*T_c_* and −Δ*S_m_^peak^*) of the full AM, and the mechanism involved, were studied.

## 2. Materials and Methods

Samples with the nominal composition of Gd_45_Co_50−x_Fe_x_Al_5_ (x = 2, 5, 10) were prepared by melting the mixture of pure Gd, Co, Fe and Al (purity > 99.9 at.%) metals in an arc furnace protected with high-purity argon. Then, the amorphous ribbons were produced by the single roller melt-spinning method with a spinning tangent speed of about 45 m/s. The ribbons with an average thickness about 35 μm were selected as the standard samples for the structure and magnetic test. The amorphous properties were characterized by a X-ray diffractometer with Cu Kα radiation (XRD, PANalytical, Almelo, Holland). The microstructure of the ribbon was observed on a JEM-2010F high resolution electron microscope (HREM) after electrolytic polishing the specimen under the protection of liquid nitrogen (HREM, model JEM-2010F, Tokyo, Japan). The magnetic features, such as Curie temperature and magnetization curves, were performed by the Physical Property Measurement System (PPMS, San Diego, CA, USA) 6000 of Quantum Design.

## 3. Results and Discussion

The XRD patterns of the Gd_45_Co_50−x_Fe_x_Al_5_ (x = 2, 5, 10) as-spun ribbons are shown in Figure 1. The XRD curves do not show significant sharp crystal peaks at the whole measured angle, and only at 2θ = 35° show the representative diffuse scattering peaks that are unique to the disordered structure. The results preliminary judge that the Gd_45_Co_50−x_Fe_x_Al_5_ (x = 2, 5, 10) ribbons are amorphous structures.

Figure 2a shows the *M*−*T* diagrams of the Gd_45_Co_50−x_Fe_x_Al_5_ (x = 2, 5, 10) amorphous ribbons under a magnetic field of 0.03 T (the solid line). Since the magnetization of the alloy decreases dramatically near the *T_c_*, the *T_c_* of these samples, that were obtained by deriving their *M*−*T* curves, as shown by the dotted line in the figure, were about 262 K, 281 K and 317 K, respectively.

The high *T_c_* of the Gd_45_Co_50−x_Fe_x_Al_5_ (x = 5, 10) AM, which is comparable to the Gd-based AM with the highest Curie temperature available to date, and also within the Curie temperature range of Fe-based AM, means that it can be applied around room temperature. If excellent MCE can be obtained, these alloys will be very promising as magnetic refrigerants. Compared to Gd_45_Co_50_Al_5_ AM, the *T_c_* of the sample increased by 3.5%, 11.1% and 25.3%, with Fe content added, from 2% to 10%. With the increase in Fe content, the *T_c_* of the alloy is obviously improved in a nonlinear increasing trend, although the proportion of TM content remains unchanged. This anomaly may be related to the stronger direct 3*d*-3*d* interactions (Fe-Fe and Fe-Co) that were introduced by the addition of Fe, because the magnetic moment of Fe (=5.4 *μ_B_*) itself is much larger than that of Co (=4.8 *μ_B_*). The effect of the magnetic moment of transition element on the Curie temperature of the alloy may be more complicated in practice. The nominal magnetic moments (*µ_nom_*.), that were calculated by simply adding the magnetic moments in atomic proportions, are about 6.01 *µ_B_*, 6.03 *µ_B_* and 6.06 *µ_B_*, respectively. Meanwhile, according to the Curie–Weiss law [24], the effective magnetic moments (*µ_eff_*) of Gd_45_Co_50−x_Fe_x_Al_5_ (x = 2, 5, 10) AM are about 8.48 *µ_B_*, 11.29 *µ_B_* and 11.75 *µ_B_* in turns, as shown in the inset curves in Figure 2a; the effective magnetic moment increases with increasing iron content. The larger effective magnetic moment indicates that the interaction of the alloy may be more complex (due to the appearance of nanocrystalline particles in the Gd_45_Co_50−x_Fe_x_Al_5_ (x = 5, 10) alloy), but the magnetic moment of the alloy is enhanced in general, which further indicates that the addition of Fe enhances the magnetic moment of the alloy, thus enhancing the *T_c_* of the Gd_45_Co_50−x_Fe_x_Al_5_ (x = 2, 5, 10) alloy.

The hysteresis loops of the Gd_45_Co_50−x_Fe_x_Al_5_ (x = 2, 5, 10) amorphous ribbons under the field of 5 T are shown in Figure 2b. The samples display ferromagnetic (FM) properties at 10 K and paramagnetic (PM) properties at 380 K. At a low temperature of 10 K, the saturation magnetization (*M_s_*) of the alloy reaches 155.5 Am^2^/kg, 154.5 Am^2^/kg and 146.2 Am^2^/kg with the Fe content from 2% to 10%, as shown in the enlarged illustration in the upper right corner of Figure 2b, and the coercivity (*H_c_*) is about 66.3 Oe for x = 2, 41.2 Oe for x = 5 and 24.8 Oe for x = 10, all of which are less than 100 Oe, as the details show in the bottom right illustration in Figure 2b. The low coercivity fields characteristic of soft magnetic materials and relatively high *M_s_* indicate that the alloy can be easily magnetized and demagnetized, which is very beneficial to the acquisition of reversible MCE and improvements in refrigeration efficiency.

The isothermal *M*−*H* curves of the Gd_45_Co_50−x_Fe_x_Al_5_ (x = 2, 5, 10) AM under 5 T are shown in the inset of Figure 3. At low temperatures that are under the *T_c_*, the alloys show obvious FM characteristics, and the curves have high magnetic susceptibility, which can increase rapidly and reach saturation under a small external magnetic field. Near the *T_c_*, the FM to PM transition occurs, and the magnetization decreases obviously. When the temperature exceeds the *T_c_*, the alloys are almost completely changed to PM, and the curves produce similar linear relations.

The Arrott curves (*H*/*M*−*M*^2^) of the alloy can be obtained by transforming the isothermal *M*−*H* curves, and the type of MPT in the magnetic field can be further verified by the Arrott curves in Figure 3. The slope of the Arrott curve of Gd_45_Co_50−x_Fe_x_Al_5_ (x = 2, 5, 10) AM is positive in the picture (a), (b), (c). According to the Banerjee rule [25], the slope of the curve is related to the thermal spin perturbation; that is, only the positive slope of the curve indicates that the alloy is undergoing a SOMPT. Therefore, these alloys are typical SOMPT materials, which allows them to have smaller magnetic and thermal hysteresis loss in MCE. At the same time, their wide-phase transition regions make the alloys have a continuous change in −Δ*S_m_* to produce a relatively large RC.

The temperature dependence of isothermal −Δ*S_m_* curves (−Δ*S_m_*−*T*) of the alloy can also be obtained by applying Maxwell’s equation—∆*S*_m_(*T*, *H*) = *S*_m_(*T*, *H*) − *S*_m_(*T*, 0) = ∫_0_^H^(*∂M*/*∂T*)_H_ d*H*—to the isothermal *M*−*H* curves. Figuring out the −Δ*S_m_* under 1 T–5 T field, at different temperatures of the Gd_45_Co_50−x_Fe_x_Al_5_ (x = 2, 5, 10) AM, as the curves show in Figure 4a–c, it can be seen that all the curves exhibit a broad characteristic of SOMPT around the −Δ*S_m_^peak^*. As shown in Table 1, the −Δ*S_m_^peak^* of all samples under the field of 5 T are about 4.04 J/(kg·K) for Gd_45_Co_48_Fe_2_Al_5_, 3.24 J/(kg·K) for Gd_45_Co_45_Fe_5_Al_5_ and 2.57 J/(kg·K) for Gd_45_Co_40_Fe_10_Al_5_, respectively. Another important indicator of measuring the comprehension MCE of the alloy is the refrigeration capacity (RC = −∆*S_m_^peak^* × δ*T_FWHM_*), and the RC of the Gd_45_Co_50−x_Fe_x_Al_5_ (x = 2, 5, 10) AM is calculated as 723 J/kg when x = 2, ~740 J/kg when x = 5 and ~770 J/kg when x = 10 under the field of 5 T, respectively. Compared with other results in the literature [18,19,20,21,22,23,26,27,28,29,30,31], the relatively large RC of the Gd_45_Co_50−x_Fe_x_Al_5_ (x = 2, 5, 10) AM are comparable to that of the Gd-Co binary AM (~720 J/kg), and are obviously better than that of the Fe-Zr-B-based AM (~600 J/kg). They are even superior than those of Gd_50_Co_50−x_Fe_x_ (x = 2, 5) (~700 J/kg) and Co_50_Gd_50−x_(Fe/Ni)_x_ (x = 1, 2, 3) (~680 J/kg) AM with the same Curie temperature. The RC results show that the addition of Fe can obviously improve the RC of the alloys. Unfortunately, compared to the original composition of Gd_45_Co_50_Al_5_, the −Δ*S_m_^peak^* of the samples is decreased by 11.2%, 28.8% and 43.5% with Fe content from 2% to 10%, respectively. Although the highest *T_c_* among Gd-based AM is obtained in Gd_45_Co_40_Fe_10_Al_5_ alloy, and its comprehensive RC is improved due to its wider temperature variation range, its magnetic entropy change property is seriously affected.

This phenomenon may be related to the variation tendency of *M*−*T* and −Δ*S_m_*−*T* curves. Compared with x = 2%, it is not hard to see, from the *M*−*T* curves, that the changes are more slow at Curie temperature when x = 5% and 10%, which is more conducive to the alloy to obtain larger RC. Furthermore, it can also be found from the curve of Gd_45_Co_40_Fe_10_Al_5_ that the −Δ*S_m_*−*T* curve of the alloy displays a nearly platform-like feature, and the peak position of its −Δ*S_m_* is shifted, which is different from that of other typical amorphous alloys. By constructing the −Δ*S_m_^peak^*~*T_c_*^−2/3^ curve and their linear fitting, as shown in Figure 5a, it is easy to find that Gd_45_Co_50−x_Fe_x_Al_5_ (x = 0, 2, 5) AM follows the −Δ*S_m_^peak^*~*T_c_*^−2/3^ relationship [32]. Meanwhile, Gd_45_Co_40_Fe_10_Al_5_ deviates from the *ln*(−Δ*S_m_^peak^*)~*ln*(*T_c_*) path, which may be caused by the presence of some heterogeneous structures—the nanoclusters—in the alloy that maintain the magnetization and do not decrease significantly at the Curie temperature of the alloy.

This can be further explained by the field dependence of the −Δ*S_m_* characteristics of the alloy. The −Δ*S_m_*∝*H^n^* relationship was established according to the Arrott–Noakes equation [33,34], as shown in Figure 5b. For the SOMPT materials, especially for typical amorphous alloys, the variation of exponent *n* is roughly as follows: under the *T_c_* region, the alloy exhibits ferromagnetism, and the exponent *n* is close to 1 and decreases with increasing temperature around the *T_c_* region; then, the exponential *n* is gradually closer to 2 with the temperature reaching the fully paramagnetism state of the alloy. In the mean field theory, the exponent *n* near *T_c_* is roughly 0.67; however, due to the inhomogeneity structure in practice, which may be due to the existence of atomic clusters, such as amorphous matter, free volume and even nanocrystalline and so on, the exponent *n* is roughly between 0.75 and 0.78 in the amorphous alloy. As can be seen in the linear fitting of the *ln*(−Δ*S_m_^peak^*) and *ln*(*H*) in the inset of Figure 5, Gd_45_Co_48_Fe_2_Al_5_ AM fits well with the actual situation, indicating the fully amorphous structure of the alloy. Meanwhile, in the Gd_45_Co_50−x_Fe_x_Al_5_ (x = 5, 10) alloy, the exponent *n* reaches 0.833 and 0.905, respectively, which indicates that the nanocrystals exist in the alloys of these two components [35,36,37,38,39]. As shown in Figure 5c,d, the HRTEM image of the Gd_45_Co_45_Fe_5_Al_5_ alloy showed a small amount of clusters, with a size limit below 5 nm, embedded in the disordered matrix, while the Gd_45_Co_40_Fe_10_Al_5_ alloy showed local nanocrystallization in the amorphous matrix, and the diameter of the cluster is slightly larger than 5 nm. Previous results have shown that the presence of short range order [37,38], with the limits of 5 nm, will lead to an slight increase in exponential *n* around 0.8. This is beneficial to the improvement in −Δ*S_m_*, while the appearance of the intermediate range order is not conducive to the improvement in the −Δ*S_m_*. Therefore, the obvious deterioration of −Δ*S_m_* of Gd_45_Co_40_Fe_10_Al_5_ in this paper may be related to the coarsening of nanocrystals in the structure. Combined with the above results, only the Gd_45_Co_45_Fe_5_Al_5_ alloy shows excellent comprehensive magnetic properties with *T_c_* of about 281 K, −Δ*S_m_^peak^* of about 3.24 J/(kg·K) at 5 T and 1.49 J/(kg·K) under 2 T, and RC of nearly 740 J/kg at 5 T and about 300 J/kg under 2 T, respectively. The relatively excellent magnetic performance of the composite Gd_45_Co_45_Fe_5_Al_5_ amorphous alloy can provide a good basic alloy and experimental guidance for further improving the magnetic entropy change performance of a Gd-based amorphous alloy.

## 4. Conclusions

In conclusion, by the addition of Fe to the base component of the Gd_45_Co_50_Al_5_ ternary alloy, the Gd_45_Co_50−x_Fe_x_Al_5_ (x = 2, 5, 10) amorphous alloys, with an average thickness of about 35 μm, were prepared, and the effect of Fe addition on the Curie temperature and magnetic entropy change proprieties of the alloys, as well as the mechanism involved, were systematically studied. The XRD results show that all the prepared alloys are amorphous. The addition of Fe significantly improved the *T_c_* of the alloy from 253 K to 262 K, 281 K and 317 K, corresponding to the Fe content increasing from 0% to 2%, 5% and 10%, respectively. This was mainly caused by the introduction of stronger 3*d*-3*d* direct interactions between the TM atoms with the addition of Fe, so that the *µ_eff_* of the alloy increased gradually from 8.48 *µ_B_* to 11.29 *µ_B_* and 11.75 *µ_B_* with the increase in Fe content and the *T_c_* was obviously enhanced. The alloys also showed good soft magnetic proprieties at low temperature, and the coercivity was negligible. The similar Arrott plots and typical magnetocaloric behaviors indicated the SOMPT feature of the alloy, and the −Δ*S_m_*–*T* curve of the alloy showed a broader curve with the increase in Fe content. The relation of −Δ*S_m_^peak^*~*T_c_*^−2/3^ deviated from the linear relationship and the −Δ*S_m_*∝*H^n^* relationship was slightly different from that of other amorphous alloys, with the exponential *n* values being about 0.771, 0.833 and 0.905 around the *T_c_*. This abnormal linearity was mainly associated with the intermediate range order nanocrystals (usually larger than 5 nm) in Gd_45_Co_40_Fe_10_Al_5_. The intermediate range order caused the magnetization properties of the alloy to be affected near the *T_c_*, thus deviating from these empirical formulas. The Gd_45_Co_40_Fe_10_Al_5_ alloy with short range order showed better magnetic properties with *T_c_* of about 281 K, −Δ*S_m_^peak^* about 3.24 J/(kg·K) and RC of nearly 740 J/kg under the field of 5 T. These results are expected to provide the foundation and guidance for further improving the room temperature MCE of Gd-based amorphous alloys.

## Figures and Tables

**Figure 1 materials-16-04571-f001:**
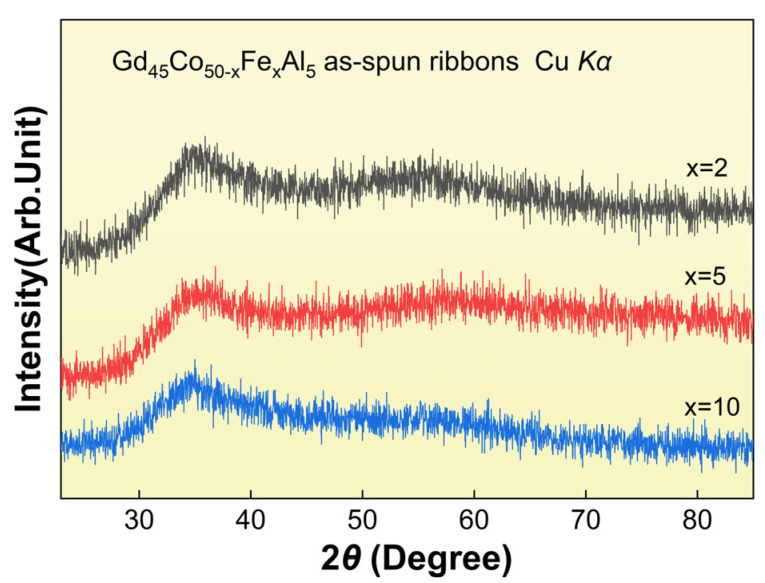
XRD patterns of the Gd_45_Co_50−x_Fe_x_Al_5_ (x = 2, 5, 10) as-spun ribbons.

**Figure 2 materials-16-04571-f002:**
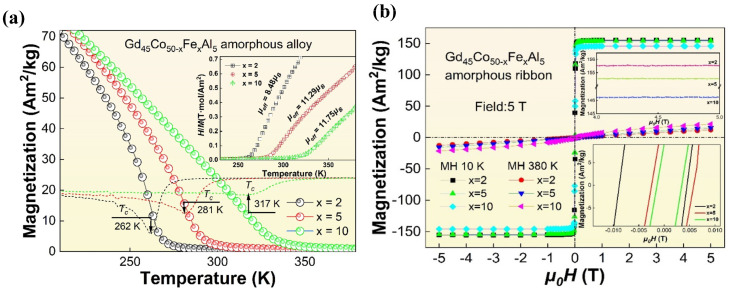
(**a**) *M*−*T* curves of Gd_45_Co_50−x_FexAl_5_ (x = 2, 5, 10) amorphous alloys under a magnetic field of 0.03 T (the solid line) and the corresponding d*M*−d*T* curves, that were obtained by taking the derivative of *M*−*T* curves, as shown by the dotted line in the figure; the inset is *μ_eff_* of the amorphous alloys, illustrated by (*H*/*M*−*T*) curve. (**b**) The hysteresis loops of the Gd_45_Co_50−x_FexAl_5_(x = 2, 5, 10) amorphous alloys measured at 10 K and 380 K under a field of 5 T. The saturation magnetization (*M_s_*) and coercivity (*H_c_*) details are shown in the illustration (**b**).

**Figure 3 materials-16-04571-f003:**
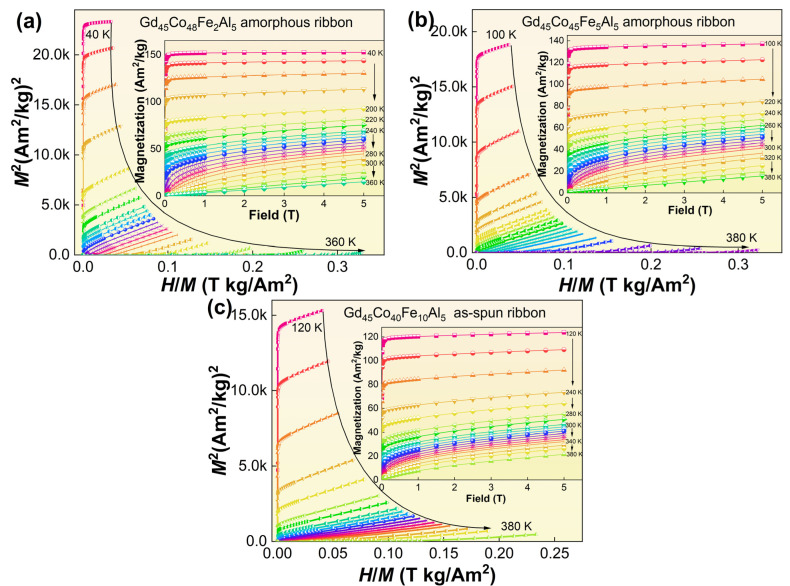
The Arrott curve of the (**a**) Gd_45_Co_48_Fe_2_Al_5_, (**b**) Gd_45_Co_45_Fe_5_Al_5_ and (**c**) Gd_45_Co_40_Fe_10_Al_5_ amorphous alloys. The insets are the *M*−*H* curves at different temperatures under adiabatic conditions.

**Figure 4 materials-16-04571-f004:**
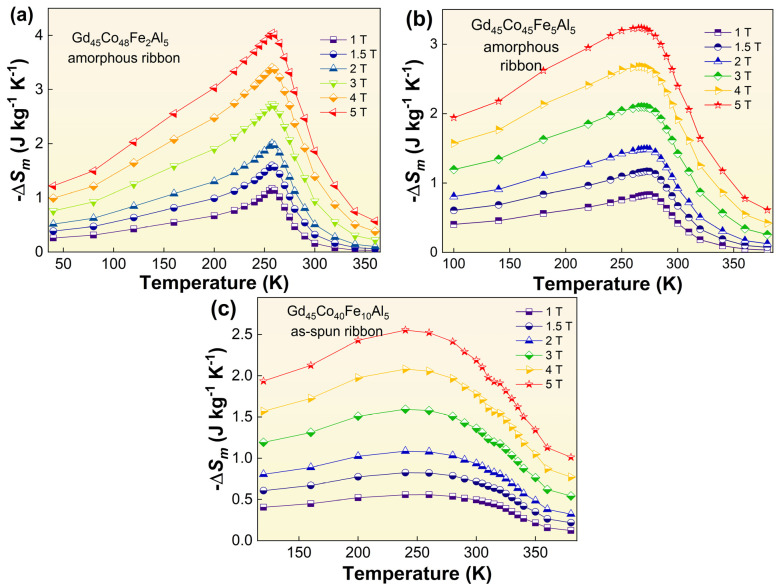
The −Δ*S_m_*−*T* curve of the (**a**) Gd_45_Co_48_Fe_2_Al_5_, (**b**) Gd_45_Co_45_Fe_5_Al_5_ and (**c**) Gd_45_Co_40_Fe_10_Al_5_ amorphous alloys under 1 T–5 T field at different temperatures.

**Figure 5 materials-16-04571-f005:**
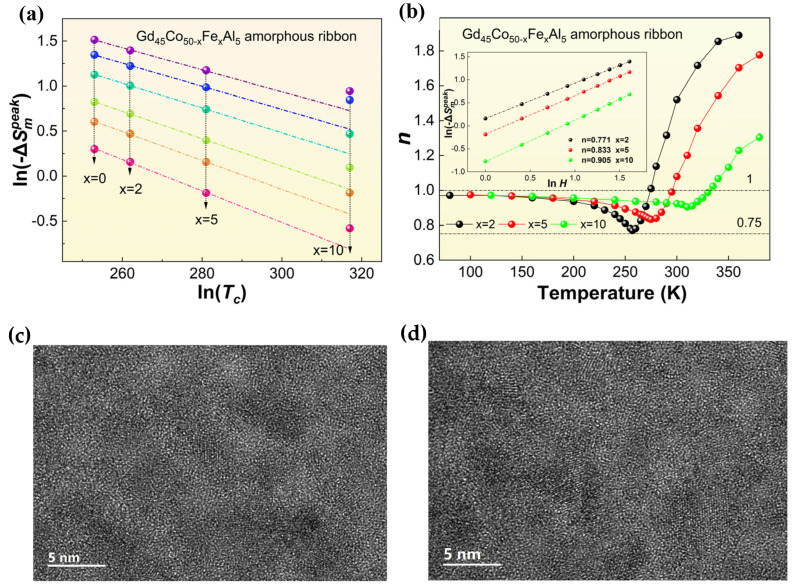
The −Δ*S_m_^peak^*~*T_c_*^−2/3^ curve and their linear fitting of (**a**) Gd_45_Co_50−x_Fe_x_Al_5_ (x = 0, 2, 5, 10) amorphous alloys; (**b**) the *n*-*T* curve of Gd_45_Co_50−x_Fe_x_Al_5_ (x = 2, 5, 10) amorphous alloys, the inset is the linear fitting of the In(−Δ*S_m_^peak^*) and In(*H*); HREM image of the as-spun ribbon (**c**) Gd_45_Co_45_Fe_5_Al_5_ and (**d**) Gd_45_Co_40_Fe_10_Al_5_.

**Table 1 materials-16-04571-t001:** Curie temperature *T_c_*(K), −Δ*S_m_^peak^* (J/(kg·K)) and *RC*(J/kg) data of the Gd_45_Co_50−x_Fe_x_Al_5_ (x = 0, 2, 5, 10) amorphous alloy and other Gd-Co-based or Fe-Zr-B-based amorphous samples with similar *T_c_*.

Samples	*T_c_* (K)	−Δ*S_m_^peak^* * (J/(kg·K))						*RC* (J/kg)		Ref.
		1 T	1.5 T	2 T	3 T	4 T	5 T	2 T	5 T	
Gd_45_Co_50_Al_5_	253	1.35	1.83	2.28	3.09	3.85	4.55	260	728	Present work
Gd_45_Co_48_Fe_2_Al_5_	262	1.17	1.60	2.00	2.73	3.40	4.04	274	723	
Gd_45_Co_45_Fe_5_Al_5_	281	0.83	1.17	1.49	2.10	2.68	3.24	~300	~740	
Gd_45_Co_40_Fe_10_Al_5_	317	0.56	0.83	1.10	1.60	2.33	2.57	~310	~770	
Gd_50_Co_50_	267	1.38		2.36	3.16	3.92	4.6	212.8	685.9	[18]
Gd_50_Co_48_Fe_2_	277			2.24			4.44		~700	[21]
Co_50_Gd_49_Fe_1_	283			2.32			4.4	226.7	680	[22]
Co_50_Gd_48_Fe_2_	297			1.98			3.96	~241	~674	
Co_50_Gd_49_Ni_1_	273			2.53			4.90	263.2	684.6	[23]
Co_50_Gd_48_Ni_2_	280			2.26			4.46	239.3	685.8	
Co_50_Gd_47_Fe_2_Ni_1_	302			1.85			3.73	~211	~620	[31]
Co_50_Gd_46_Fe_2_Ni_2_	310			1.59			3.28	~231	~670	
Fe_88_Zr_8_B_4_	291			1.5			2.81	200	551	[27]
Fe_87_Co_1_Zr_8_B_4_	317			1.61			3.24		~600	[28]
Fe_87_Zr_8_B_4_Sm_1_	308			1.65			3.27		>550	[29]

* The maximum magnetic entropy change (−Δ*S_m_*) value in the −Δ*S_m_*–*T* curves.

## Data Availability

The data of this study are available from the corresponding author upon reasonable request.

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
