# Peer review of "The Effect of Fe Addition on the Curie Temperature and Magnetic Entropy of the Gd45Co50Al5 Amorphous Alloy"

_materials, 2023, doi:10.3390/ma16134571_

Round 1

Reviewer 1 Report

This article presents and discusses interesting results on the magnetic entropy of GfCoAl with Fe addiitions. In general, the manuscript is well organized and properly explained. However, further clarifications are required before considering for publication, as indicated below

 1.You stated ate the Abstract that you will report the effect of Fe addition on the glass forming ability, but this property is not mentioned within main text. Please, correct this inconsistency.

2.Within “Materials and Methods” section, please indicate the roll speed used for the melt-spinning process.

3.XRD patterns look noisy. Patterns with improved resolution are strongly suggested.

4. Transmission Electron Microscopy studies for your alloys are strongly suggested in order to provide evidence of formation of nanocrystals or atom clusters and hence, correlate with results shown in Fig. 5. Determination of Tc for each sample is not really specified. The “dramatic change of magnetization” that you mentioned is related with a change of slope of the curve M-T?. Please, be more specific on the way you determine Tc.

5. ForFig.2a, What was the minimum temperature you measured?. It would be interesting to show M-T plot well below 200 K. Please, make a comment on this.

6.For Fig.2b, if you use mks unis for Magnetization, the appropriate units for external field H are “A/m”. Please, correct units in Fig.2b and label “Field H”.

7.How do you calculate the “nominal magnetic moment”?. Why is so different this “nominal magnetic moment” compared with the “effective magnetic moment”?

8.Hysteresis loops at 10 K exhibit a decreasing tendency for saturation magnetization with increasing Fe content, which represents the opposite behavior showed by the effective moment tendency. How do you explain this inconsistency?.

9. Coercivity fields between 24 and 66 Oe are far of being “approximately negligible”. You should state such values as “low coercivity fields characteristic of soft magnetic materials”. Please, make a comment on this.

Author Response

Dear reviewers

Thank you for your letter dated 3 June 2023. We were pleased to know that our work was rated as potentially acceptable for publication in Journal, subject to adequate revision. We thank the reviewers for the time and effort that put into reviewing the previous version of the manuscript. These suggestions have enabled us to improve our work. Based on the instructions provided in your letter, we have revised the article accordingly.

The comments are reproduced and our responses are given directly afterward in a different color (blue). we hope these responses can clarify the problems in the paper.

Sincerely

Benzhen Tang

Reviewer 2 Report

In the manuscript, magnetic properties and magnetocaloric effect have been studied for Gd45Co50-xFexAl5 amorphous alloys. Aluminum alloying makes the amorphization more easily, while Fe substitution for Co leads to increasing Curie temperature. The authors found that the isothermal magnetic entropy change decreases with the Fe addition, while the relative cooling power increases. The obtained materials show a smaller MCE than pure Gd. Nevertheless, the results are new and can be published in Materials. However, some parts of the manuscript required major revision.

1. A part of the text in which the magnetic moments are discussed, should be presented more clearly. What is the value of magnetic moment of Fe and Co? In metallic state, it should be considerably smaller than 5.4 mB and 4.8 mB, which is an effective moment in paramagnetic state. What is the nominal magnetic moment? Do you take into account the Gd moment? Please provide formula for its calculation. If increase in the Curie temperature is associated with “enhancing the magnetic moment of the alloy” with Fe addition, then why magnetization at 10 K decreases?

2. The Curie temperature of the samples was “obtained from the dramatic change of magnetization in M-T curves”. It will be much better to use some physical law to detect the Tc value, for example, Arrott plots of Fig. 3.

3. In Abstract and in Introduction the value of the maximum magnetic entropy change should be accompanied by the field change.

4. In Fig. 2a, the label ‘Field applied: 300 Oe’ should be omitted, since all the fields in the manuscript are given in Tesla.

English is understandable, however, minor editing of English language required

Author Response

(The authors gave the same response as above.)

Round 2

Reviewer 1 Report

Authors have addressed properly all questions raised previosuly.

Reviewer 2 Report

The authors answered all the questions. The manuscript can be published in its present form.